# The Use of Neurofeedback in Sports Training: Systematic Review

**DOI:** 10.3390/brainsci13040660

**Published:** 2023-04-14

**Authors:** Łukasz Rydzik, Wojciech Wąsacz, Tadeusz Ambroży, Norollah Javdaneh, Karolina Brydak, Marta Kopańska

**Affiliations:** 1Institute of Sports Sciences, University of Physical Education, 31-571 Krakow, Poland; 2Department of Biomechanics and Sports Injuries, Kharazmi University of Tehran, Tehran 14911-15719, Iran; 3Student Science Club “Reh-Tech”, Institute of Medical Sciences, University of Rzeszów, 35-959 Rzeszow, Poland; 4Department of Pathophysiology, Institute of Medical Sciences, Medical College of Rzeszów University, 35-959 Rzeszow, Poland

**Keywords:** biofeedback, neurofeedback, sport, training, performance

## Abstract

Biofeedback training is a method commonly used in various fields of life, for example, in medicine, sports training or business. In recent studies, it has been shown that biofeedback, and neurofeedback, can affect the performance of professional athletes. Training based on the neurofeedback method includes exercising the brain waves. The aim of the article is to evaluate the influence of neurofeedback training on the physical fitness of professional athletes representing various sports disciplines, such as judo, volleyball and soccer. Based on 10 scientific papers from various sources, including PubMed, the latest research on neurofeedback and its impact on athletes has been reviewed. On the basis of the literature review from 2012 to 2022 on the neurofeedback method in sports training, it can be stated that this type of practice has a significant impact on physical fitness and sports performance. This review comprised 10 research studies with 491 participants in the neurofeedback groups, and 62 participants in the control group. Two reviewers independently extracted data and evaluated the quality of the studies utilising the PEDro scale. Properly planned and conducted neurofeedback training affects stimulation and improvement of many variables (reducing stress levels, increasing the ability to self-control physiological factors, enhancing behavioural efficiency and meliorating the speed of reaction to a stimulus).

## 1. Introduction

Sports training is a specific form of exercise. Its effectiveness depends on properly selected methods. Excessive training loads can result in overtraining, while insufficient training loads do not lead to significant sports achievements [1]. It should be pointed out that all professional athletes require not only basic physical training to improve motor skills, but also mental training, which develops athletes’ concentration and resistance to stress [2]. The methods used in sport psychology include rest management, psychological assessment and neurofeedback [3]. Over the past decade, research teams from around the world have focused on studies regarding the effectiveness of EEG (electroencephalogram) training and its benefits for athletes. Nowadays, achieving sports performance at the highest level requires not only motor preparation, but also taking mental training into consideration. Research in this area allows proving the benefits for the psychophysiological sphere of subjects such as stress control, which is a must in professional sports. Pressure related to results, training staff and sponsors are factors that interfere with the management of stress, directly affecting performance and athletic ability. In our review, we aim to gather scientific evidence on the effectiveness of biofeedback training and to determine its benefits for athletes.

Biofeedback is a brainwave training process. These waves are categorised on the basis of their frequency. Neurofeedback (NFB) is a method that assists subjects in consciously controlling their brain waves. In fact, EEG is recorded during the neurofeedback treatment [4]. Neurofeedback therapy utilises brain waves as feedback information [5]. In neurofeedback therapy, data are collected using EEG, and the principles of biological control theory are used to obtain feedback regarding psychological and physiological processes [6]. The authors emphasise the fact that research on this topic is necessary because these divisions are conventional and not all is certain, while conducting more targeted therapies requires a more thorough exploration of wave knowledge. Depending on the frequency of the waves, we can distinguish alpha waves (8–12 Hz) that are related to the relaxation state when the eyes are closed, beta waves (above 13 Hz) that correspond to concentration, theta waves (4–8 Hz) that cause increases over temporal sites during encoding, maintenance and retrieval, and finally, delta waves (0.5–3.5 Hz) experienced by humans during sleep [7]. The function of biofeedback is given in Figure 1. The 1960s and 1970s were the periods in which the scientific method of EEG training was first demonstrated. The purpose of this was to increase alpha wave activity, which would translate into an increased degree of relaxation. By distinguishing different frequencies and their roles, we can assign specific states to the appropriate waves associated, in the case of gamma waves, with intense concentration of attention and assisting the brain in processing, while combining information from different areas of the brain. Multi-tasking during athletic competition requires athletes to maintain a maximum level of focus for as long as possible [8]. Neurofeedback enables better control over the physiological processes taking place in the human body, which are difficult to control [4]. It is a method widely used all over the world and its area of occurrence is broad. Biofeedback treatments are used in medicine, where they have become a substitute for many pharmaceuticals [8]. Moreover, biofeedback can stimulate the proper emotional development of children and adolescents [9]. The same method is also common in geriatric treatments [10]. These patients often suffer from various types of cognitive impairments resulting from damage to the central nervous system [4]. EEG-NFB training could be used according to the FITT principles (frequency, intensity, time and type) independent of age and fitness level [11]. Depending on the need, neurofeedback is a safe and effective method that brings the desired effect and is suitable for all age groups [12]. For this reason, the neurofeedback method becomes even more important and requires constant improvement. This innovative system proves its effectiveness not only in treating patients with diseases, but also in other areas of life, such as music, business, acting and sports [9].

It is important to focus on the role of neurofeedback in professional sports. In the 1980s, scientists developed a method of periodisation for sports training [13]. There are three basic periods in professional training, accordingly: preparation period—developing fitness and training, starting period—maintaining a high level of performance, and transition period—physical and psychic recovery. In each of these phases, the athlete’s training is based not only on the physical aspect, but also on the mental one. Neurofeedback is an approach, the proper application of which allows for cooperation of the physical and mental spheres [13]. The concept of electrical activity through electroencephalography (EEG) neurofeedback has a wide scope. The commonly used concept of “biofeedback” should be associated mainly with the measurement and evaluation of various body parameters, associated with special electronic devices [4]. Neurofeedback training has the function of stimulating the athlete’s body (heart rate and respiration) to self-regulate and create an autonomous impulse that can help in taking appropriate actions and making proper decisions at key moments in a competition [13]. Neurofeedback training properly adjusted to the athlete’s individual abilities has an impact on psychophysiological consistency [14]. Achieving this state causes athletes to feel positive emotions, and their heart rhythm, as well as their level of perceived stress, is constantly more easily controlled [15].

In research on the subject, evidence has been demonstrated regarding the effectiveness of neurofeedback training as a mental support method for professional athletes. An important aspect is the proper selection of training protocols and specialised staff, which allows influencing the desired parameters [12]. In general, in NF applications, brain signals are recorded with different neuroscientific methods [electroencephalography (EEG), magnetoencephalography (MEG), functional magnetic resonance imaging (fMRI), near-infrared spectroscopy (NIRS)], processed in real time by a computer and fed back to the NF user online via visual, auditory and/or tactile feedback. Motor imagery of movements is used as a mental strategy in neurofeedback applications to gain voluntary control over activity in motor areas of the brain. Motor imagery, defined as mental simulation of a kinaesthetic movement [16], can also modulate activities in the sensorimotor cortex without any physical movements of the body. It is often used as a mental strategy in NF applications [17]. The MI recording technique captures signals based on a user’s imagination of performing a specific task, for example, limb movement but without actually moving the limb. The imagination of moving a unilateral limb causes variation of activations in a specific cortex area [18], which are further translated into electrical signals propagated by volume conduction through multiple brain tissue. Motor imagery has been used to enhance athletic performance [19].

In 2010, a study was also conducted among Canadian athletes participating in the Vancouver Olympics, which showed that the use of NFB increased their stress control, which resulted in better results at the Olympics compared to the previous year [20]. In a review study, Mirifar et al. examined protocols and neurofeedback applications, as well as their effect on sports performance. The search process was finalised on 30 June 2016. In this study, it was shown that, so far, the majority of published studies support the statement that NFT effectively improves athletes’ performance in a specific sports task and/or in relevant underlying aspects of cognition and effect. In the end, they stated that the final conclusion regarding the validity of the findings in this review study is quite different from the positive conclusions drawn in previous studies. More research efforts need to be made in the field of sports to uncover constraints and specifications for NFT in sports. Thus, despite some indications that NFT use is effective for improving sports performance, substantial evidence for its effectiveness is missing [6].

The aim of the paper is to review the scientific literature on the impact of neurofeedback methods used in sports training on professional athletes’ performance. In our work, we attempt to answer the questions as to what neurofeedback training protocols are effective for achieving appropriately targeted goals in athletes and how and to what extent they shape their mental abilities.

## 2. Materials and Methods

This study conforms to the Preferred Reporting Items for Systematic Reviews and Meta-Analyses (PRISMA) statement [21] and the Cochrane Collaboration Handbook for Systematic Reviews of Interventions [22].

The review of articles was carried out by searching for information on biofeedback and neurofeedback regarding the performance of professional athletes, using PubMed and other available scientific databases. The keywords neurofeedback in sports training and efficiency of EEG biofeedback were used to search for scientific articles indicating the impact of biofeedback and neurofeedback on sports performance. The presence of a control and research group is a desirable phenomenon in the researched subjects. It is important to have the most extensive research group, the results of which can be compared before and after training. In the review of the literature, many sports disciplines mentioned the impact of the above technique.

The purpose of the analysis was to search electronic databases with scientific articles, including PubMed, Google Scholar, Scopus, Web of Science and current scientific databases. The following keywords were used in the search: EEG, neurofeedback, sports. Publications from 2012 onwards were taken into account. In Table 1, subsequent stages of the literature search are presented. The inclusion criteria for the literature search are also given.

All research papers were included in this study. Conference proceedings, commentaries, reviews, editorials, research notes, letters, duplicated publications and expert experiences were excluded.

The following inclusion criteria were taken into account when selecting the appropriate items: The original publications included studies on athletes in which the training protocol is shown over time. The publications in which the athletes were professionally active during the protocol were considered for analysis. Age was not a criterion for exclusion, but automatically taking into account only active people, the range was approximately 18–35 years. Age was a criterion for inclusion, and considering only active people, it was within the range of approximately 18–45 years. The older range was rejected due to the existence of diseases affecting the results. Exclusion criteria included the presence of mental and physical disorders that significantly influenced the course of the study (Table 2).

### Assessment of Study Quality

The PEDro scale was utilised to examine the quality of the included studies [23]. This scale includes 11 criteria: eligibility, random allocation, allocation concealment, baseline comparability, patient blinding, therapist blinding, assessor blinding, less than 85% dropouts, intention to treat analysis, between-group comparisons, and point and variance measures [23]. Apart from the first criterion, each item scores 1 or 0 points. The maximum possible score is 10. A cut-off score of ≥6 was considered for high-quality studies [23].

## 3. Results

First, 450 studies were identified. After removing duplicates, 405 (n) remaining trials were screened for eligibility. Then, 140 studies were excluded because they were not about neurofeedback in sports, focused on unrelated subject areas (n = 120) or did not have an available full text (n = 20). Finally, according to our inclusion/exclusion criteria, 10 studies were considered acceptable for this literature review (Figure 2).

For the purpose of a detailed analysis of scientific publications, the description of the research group, training methods, duration of the research and observed results were indicated by the authors of the paper. The majority of publications covered a control and research group. In most of the studies, the effect of a specific neurofeedback method on the level of physical fitness was considered. In all of the analysed studies, it was shown that neurofeedback influences the physical and mental fitness of NFB participants. This was confirmed by controlled tests carried out before and after special training (Table 3).

From the analysed literature on the subject, high research quality was found for five scientific reports, one of moderate, and four of low risk. Table 4).

## 4. Discussion

Taking the above literature into account, it should be borne in mind that neurofeedback is a relatively new method, and its impact on various fields (such as sports) has not been well-studied by specialists. Nonetheless, neurofeedback is used in many fields as a form of treatment or training [7]. The benefits of using this method have been documented in studies worldwide [6,7,20,32,33].

### 4.1. Heart Rate, Stress, Blood Pressure and Neurofeedback

Raymond demonstrated positive changes in the heart rate regulation process among professional dancers after neurofeedback training [32]. Similar effects and increased stress resistance were also noted in studies conducted on golfers in 2012 [33]. Furthermore, it has been shown that this type of training, in addition to helping cope with stress, also affects concentration. In many cases, biofeedback and neurofeedback interventions have contributed to tremendous success [34]. In these studies, improved performance and reduced stress in athletes have been indicated. This was achieved through the combination of electroencephalogram (EEG) interventions and heart rate variability (HRV) feedback [34].

In tennis, there are various pressure situations that players experience. In a study by Habay et al., it was shown how mental fatigue (MF), a state that lowers physical and cognitive performance, affects table tennis players [35]. The results suggested a negative effect in the form of longer reaction times to inhibitory stimuli, while spectral analysis revealed increased desynchronisation of brain regions. This expands knowledge on the influence of MF on visuomotor performance. The authors suggested that both table tennis coaches and other professionals should be aware of and prevent the effects of MF. Future research should be directed towards this goal. Both novice and experienced table tennis players could play better if they reduced stress and pressure during the game. When stress and negative emotions take over, technical and tactical errors are more frequent. Therefore, three pressure-reducing strategies were proposed: implicit or errorless learning (learning skills without explicit knowledge), self-awareness training (being monitored and evaluated), and perceptual-motor task training with mild anxiety [35]. Meanwhile, in a study by Pineda-Hernández, the author aimed to monitor activation while imagining a neutral (NSI) and pressure situation (PSI) based on the analysis of brain waves, heart rate and subjective evaluation of athletes [36]. Monitoring of activation while imagining a pressure situation showed an increase in heart rate frequency, as well as approximate and sample entropy, while a decrease in gamma wave activity was observed compared to initial and final concentrations on breathing, which was better compared to imagining a neutral situation. However, gamma waves increased at the moment of maximal pressure [35,36].

### 4.2. Injuries and Rehabilitation

In a study conducted among professional Serie A football players (the highest league in Italy), Esteves et al. demonstrated that neurofeedback training plays an important role in preventing injuries [37]. The neurofeedback method not only allows athletes to achieve high performance, but also influences injury prevention and rehabilitation processes. Neurofeedback permits analysis of the load exerted on both limbs during squats and landings by athletes. This method enables the planning of training for these athletes in preventing knee injuries, especially anterior cruciate ligament (ACL) tears in the knee [38]. The use of neurofeedback as a way of rehabilitating athletes after injuries was demonstrated in the research by Alahakone and Senanayake. They showed that momentary feedback received through vibrational stimuli can reduce postural sway based on measurements of trunk inclination angle [39]. This phenomenon allows for central stabilisation. In the publications by Malik and Senanayake, the crucial role of this method in monitoring the health status of athletes in the rehabilitation process is highlighted. Rapid feedback allows physicians and physiotherapists to make quick decisions regarding the rehabilitation of athletes, thus, allowing for a faster return to full functionality [40]. This method can be used in the rehabilitation of individuals with treatment-resistant sports-related injuries [41]. Perry et al. also drew attention to increased effectiveness and improved self-regulation with the support of neurofeedback [42]. Malik and Senanayake demonstrated the effectiveness of this method in the rehabilitation of a specific injury, namely ACL tears. The use of neuromuscular signals permits the analysis of movement in athletes affected by injury, therefore enabling physicians to take immediate action [43].

### 4.3. Music as a Form of Training

There are many unconventional forms of neurofeedback considering physical aspects. Music belongs to one of these branches. In this way, it contributed to maintaining motivation in the rehabilitation process of the world champion in kayaking with chronic fatigue syndrome. Through repeated movements to musical rhythm, this outstanding kayaker was able to endure the tedious rehabilitation process and return to high psychomotor performance [44].

In the research by Maszczyk et al., it is clearly indicated that this type of training significantly affects the dynamic balance of judokas [45]. Neurofeedback training has also been suggested as a way to expand self-regulation management. This method is recommended for musicians. It can help motor coordination in disciplines that involve synchronous movements [10]. Music as feedback allows not only to maintain motivation during rehabilitation, but also provides an opportunity to achieve better physical performance. Research conducted on 20 Caucasian 400 m runners insinuates the effectiveness of this method. A suggestive correlation was shown between special synchronous music (listened to by athletes during running) and the ability to train with greater anaerobic endurance [46].

### 4.4. Performance

Not all scientists agree that an athlete will be able to unconsciously control the processes occurring in the body without electrical devices that accompany them during training. Nonetheless, neurofeedback training is being incorporated to enhance performance in sports, cognitive fields and the artistic industry. In research on the subject, it has been shown that the use of neurofeedback improves performance [47]. In a study conducted by Chen et al., the impact of auditory neurofeedback on the execution of a boxing punch was examined in professional and amateur boxers. The results confirmed the positive influence of externally directed neurofeedback on biomechanical variables related to technique and performance. Participants used additional neurofeedback information to enhance natural touch and sound signals associated with striking the punching bag. The effect of biological feedback can also be used to improve movement patterns to avoid injury, with rapid improvement in results for beginners and movement consistency in various sports [48].

Van den Berghe et al. presented a method for detecting changes in peak tibial acceleration during adaptation to low-impact self-running. These authors also used auditory neurofeedback in their study. Ten runners, with high tibial acceleration, were equipped with a wearable auditory neurofeedback system. All participants found a way to run with lower peak tibial acceleration. No change points were detected in the absence of neurofeedback. For each person in the neurofeedback conditions, at least 1 change point was detected, indicating that runners responded quickly to real-time auditory feedback. The system allows running with less impact load, reducing peak tibial acceleration for ground running compared to running without the device. The neurofeedback system and its strong sensory–motor coupling assisted rapid reduction of axial peak tibial acceleration during the initiation of gait retraction [49].

Furthermore, Jeunet et al. conducted a study aimed at identifying the neurophysiological (EEG) correlates of covert visuospatial attention (CVSA) that could be used in NF training procedures dedicated to improving the performance of football goalkeepers. A significant positive correlation was revealed between the improvement of CVSA ability in athletes and an increase in their resting alpha power. This result suggests the possibility of designing innovative ecological training procedures for goalkeepers, such as using a combination of NF and cognitive tasks performed in virtual reality [15].

In a study conducted by Mikic et al., the impact of holistic training on physiological and behavioural measures in semi-professional athletes was evaluated. During the neurofeedback session, participants were accompanied by self-relaxation and audio–visual stimulation following daily physical activity [29]. The training led to an increase in alpha and beta1 power in athletes at rest with their eyes closed. The same level of power in each frequency band was observed in participants with their eyes open, which was not observed in the control group for which beta1 power in the second measurement was lower.

### 4.5. Frontal Cortex Activity and Dual-Layer Neurofeedback

Furthermore, Visser et al. aimed to investigate the activity of the frontal cortex during table tennis (TT) gameplay [50]. In the study, the theta activity of frontal areas was compared between gameplay and physical activity on a stationary bicycle. According to the authors, TT gameplay increases the activity of pre-frontal and fronto-central brain regions. The increase in theta activity in the frontal area was also associated with error monitoring (e.g., maintaining postural balance) and task and environmental unpredictability. The study revealed specific patterns of frontal cortex activity for exercise. It was observed that anterior and anterio-central theta power increased during open-skill exercises because they require continuous perceptual control and stimulus processing from the external environment. According to the authors, further research on open-skill exercises should be focused on studying the long-term neurophysiological responses that these exercises can elicit [50].

Additionally, Studnicki et al. characterised artifact removal strategies in neurofeedback data for whole-body tasks such as table tennis [51]. Using a dual-layer neurofeedback approach, noise channels give an alternative representation of motion artifacts, providing a fuller picture of the artifact induced by the cable than other sensors. The authors compared individual scalp, neck muscle and noise-matched channels, and accelerometer data from the body and head to determine the efficacy of reference signals in capturing noise. The study participants performed various tasks in table tennis. The hypothesis was that motion artifacts at the level of individual channels recorded by noise electrodes would correlate with raw electrocortical data recorded by the scalp electrodes. Depending on the type of sensor, spectral shapes differed: noise electrodes recorded higher power at low frequencies, and their slope was steeper; scalp electrodes followed a common 1/f curve but recorded an increase in alpha and theta power (most noticeable in standing baseline state). “The correlation between individual scalp channels and noise-matched electrodes increased during impact conditions compared to standing baseline conditions. The correlation between scalp and noise-matched electrodes located at the front of the head during standing baseline conditions and at the top of the head during impact conditions was slightly higher. We observed a small difference in the correlation between scalp channels and noise-matched electrodes in the four impact conditions”. The dual-layer approach was found to effectively characterise motion artifacts affecting scalp channels during dynamic sports such as table tennis [51]. In externally directed neurofeedback, the person receives feedback on the indirect external effect of their movements rather than direct internal feedback on the functioning of the body.

### 4.6. Research, Factors and Subjective Results

Larson et al. included only one professional athlete in their study, despite conducting 15-month neurofeedback training. The results may be subjective due to the evaluation of only one individual. Indian athlete—Abhinav Bindra—won a gold medal at the Olympics in Beijing. Her training included “neurofeedback” methods, which helped her achieve a “psychokinetic state” that contributed to her great triumph. However, we cannot fully confirm the effectiveness as the studies were conducted on a single individual [52]. The presence of a control and experimental group is a desirable phenomenon in trials. It is important to have the most comprehensive research group, the results of which can be compared before and after training.

Pusenjak et al. [14], Mirosław et al. [29] and Adam et al. [30] included both a research and control group in their research. The majority of the included studies were of a small sample size, and the size of the neurofeedback effect was not as significant. Another limitation in the field of sports is diversity. The studies were focused on different groups of athletes from separate sports disciplines. Hosseini et al. used a protocol appropriate for volleyball players [25]. Adapting different training sessions was significant in achieving the desired effect in a given field.

Future research should be focused on greater unification of desired research characteristics and replication of studies in a given sports discipline. In many of the studies, the influence of external factors and changes beyond neurofeedback training were not taken into account. In them, the results of neurofeedback should not only be presented, but also all changes identified. Another weakness of the methodology is the use of different training protocols in each study. Specific protocols and justifications should be created for a given sports discipline in future research. In each sample, professional athletes were enrolled, which is a positive phenomenon and should be continued in the future. The justification is that the application of the appropriate protocol in novices is significantly different from advanced groups, and the results vary significantly. Another difference in the protocols used is the time intervals and duration of training. Hosseini et al. A total of 29 conducted a single training session compared to Mikicin et al. [28], who used 20 training sessions, which is a significant difference. In the future, the optimal frequency of neurofeedback training should be considered, which will most effectively influence the results. Time constraints should not be limited, and as a result, the frequency or intensity of training should not be accelerated. Optimal and safe conditions for neurofeedback training that do not overload athletes’ physical or mental health should be established.

### 4.7. Limitations of the Study

Two researchers performed the literature search, study selection, quality assessment and data extraction. There were a small number of studies included in this review, which suggests a certain bias in the interpretation of the results. Most studies lacked a control group, making it difficult to determine whether the effects were due to neurofeedback interventions or non-specific effects. Another limitation is that different neurofeedback protocols with different waves and frequencies were used in the studies.

## 5. Conclusions

In all of the analysed scientific studies, the effect of sports training was confirmed on the basis of the EEG biofeedback method. It has been shown that biofeedback used in professional athletes’ training improves their ability to control psychophysiological factors, including GSR and HR after exposure to stress, and thus, contributes to the improvement of well-being (Table 5).

The analysis of selected literature allows emphasising the effects of using neurofeedback in sports training. It has been shown that the contraction of this method in the training of professional football, judoka, volleyball players and boxers, causes an increase in behavioural performance and in speed reaction, a decrease in blood cortisol levels and an improvement in motor coordination. These positive results, achieved thanks to the skilful use of the neurofeedback method, directly influenced the increase in the level of sports abilities of the athletes who used this methodology in their training.

## Figures and Tables

**Figure 1 brainsci-13-00660-f001:**
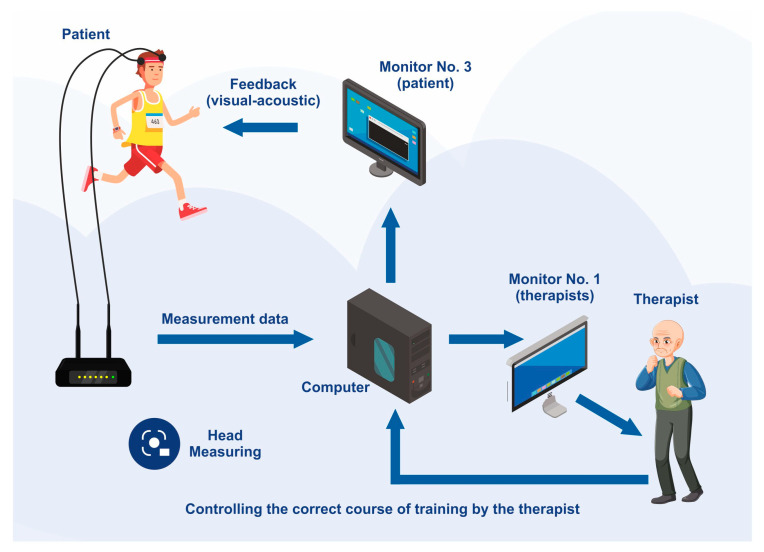
Diagram of biofeedback functioning (own diagram).

**Figure 2 brainsci-13-00660-f002:**
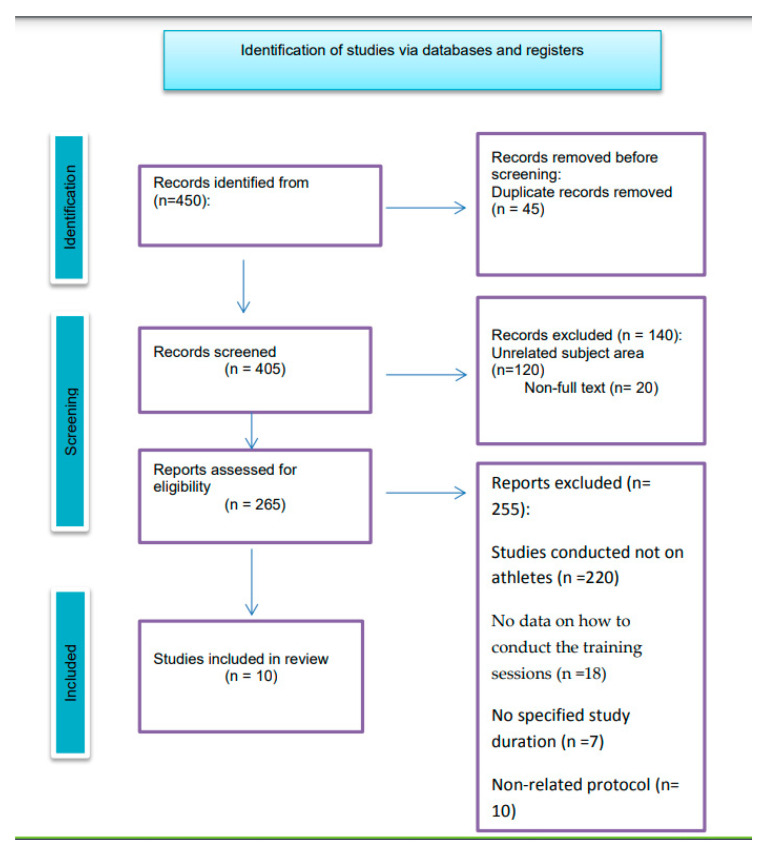
PRISMA chart for included studies.

**Table 1 brainsci-13-00660-t001:** Stages of literature search.

Search Stages	Search Phrases
1	MeSH: neurofeedback, sport, training, efficiency
2	Publications above 2012
3	Publications in English and Polish
4	Available abstract

**Table 2 brainsci-13-00660-t002:** Inclusion and exclusion criteria used for the analysis.

Inclusion Criteria	Exclusion Criteria
Publications based on research among athletes	Studies not conducted among athletes
Training based on the biofeedback method	No data on how to conduct the training sessions
Description of the training session	No specified study duration
Duration of the research	The presence of diseases affecting the results

**Table 3 brainsci-13-00660-t003:** Materials, methods and results of biofeedback training studies.

Authors and Type of Article	Experimental Group	Control Group	Neurofeedback Training	Results	Conclusions
Authors, Countries of Stud, Study Type	Gender, Age, Sample Size, Sports	Gender, Age, Sample Size, Sports	Methods, Type of Training, Protocol, Frequency of Training	Performance Level, Outcomes	Was the Training Used Effectively and How Will It Affect the Athletes?
Nika Pusenjak et al. [14], Slovenia, clinical study, a practitioner’s perspective	18 professional athletes; 4 women 14 men 16–34 years Mixed sports (archery, shooting, fencing, wake-boarding, track-and-field, volleyball, basketball, skiing, cross-country skiing, swimming, tennis, cycling, taekwondo, table tennis, carting	21 professional athletes; 4 women 17 men 16–34 years Mixed sports (archery, shooting, fencing, wake-boarding, track-and-field, volleyball, basketball, skiing, cross-country skiing, swimming, tennis, cycling, taekwondo, table tennis, carting)	BioTrace software was used in the study, which is part of the Nexus 10 (Mind Media BV, NL) biofeedback device. Parameters studied: number of breaths per minute, heart rate, blood volume and coherence, galvanic skin response (GSR) and temperature. Two tests were conducted for each subject: a short stress test (5 min) and a long stress test (14 min) for both groups. Then neurofeedback training was conducted for 1 h 2x a week for 8 weeks for the experimental group using the Wild Divine biofeedback device and the Wisdom Quest integrated software. After the training cycle, both groups were re-tested.	The researchers showed that athletes who were able to achieve psychophysiological control could achieve a state of deeper relaxation, as indicated by improved breaths per minute, lower skin conductance, lower heart rate, higher blood flow and higher breathing, as well as heart rate coherence. More athletes in the experimental group were able to control individual parameters, as opposed to the control group. Only 5% of the subjects could train heart rate control in the face of a stressful stimulus. The main difference between the 2 groups concerned the skin conductance parameter. For the short stress test, the difference in change was more than two-fold (67% to 35%). Summing all parameters, the experimental group was better by >15%.	The researchers did not base their analysis on qeeg diagnostics; however, parameters related to stress response were analysed. In order to improve the quality of the study, it would be worthwhile to continue research among a larger group of subjects and supplement it with qeeg diagnostics and brainwave recording. The studied parameters, with the exception of GSR, were inconsistent in the experimental group and it is difficult to assess the effect of training on them. In future studies, other measurement parameters should be used to assess the effect of training, but GSR shows that the neurofeedback training used in this study allowed the subjects to better control stress and psychophysiological functions in the tests used.
Camille Jeunet et al. [15], France, clinical study	17 professional athletes (1 was excluded from analysis of the results because there were problems during the EEG recording) 15 males 2 females Age 21.4 ± 5.3 years Football goalkeepers	There was no control group in the study	Two sessions were conducted, each lasting 1 h, both in the same week. Four CVSA (covert visual spatial attention) tasks were performed, and EEG activity was recorded. Failed trials were not recorded for EEG analysis. EEG data were recorded using 2 g.USBAmp amplifiers (g.tec, Austria), using 32 wet (g. tec LadyBird) scalp electrodes (F3, F1, F2, F4, FC3, FCz, FC4, C1, Cz, C2, CP5, CP3, CPz, CP4, CP6, P7, P5, P3, P1, Pz, P2, P4, P6, P8, PO7, PO3, POz, PO4, PO8, O1, Oz, O2, 10–20 system), referenced to the right ear and grounded to AFz. The α-wave band (8–14 Hz) in areas of the occipital cortex was analysed. EEG data were sampled at 512 Hz. Data were recorded using OpenViBE45 software and pre-processed using MATLAB/EEGLab 4. The study took place under conditions of isolated distance from the screen, with each subject holding their chin in a holder. The Eye Tribe eye tracker was used at a distance of 40 cm from the subject at a frequency of 60 Hz, and a 1.5° gaze deviation of visual angle from the fixation point was tolerated. In addition, the multiple object tracking (MOT) test was performed during both sessions to assess intra-session improvement. MOT exercise was implemented in Unity 5.0 using the C# language. The covert attention task was implemented in C++ as the OpenVibe45 scenario.	Athletes with improved CVSA scores also had improved MOT scores. There was no correlation between knowledge and CVSA ability in athletes.	The results of the study indicate that the use of neurofeedback training may have an effect on CVSA in goalkeepers, which may translate into athletic ability; however, this study does not allow prove the statement, it only provides the potential for its continuation. The authors highlight the shortcomings of their study and the need to work with a larger group, over a longer period of time, and using a control group. Two sessions are not enough to see clear changes in athletes’ form. Improvements in the MOT test (pre vs. post) ranged from −9.17% to 19.17%, demonstrating the potential of this type of training in athletes and its effect on CVSA.
Magdalena Krawczyk et al. [24], Poland, clinical study	6 professional sportspersons from the National Team of the Polish Judo Association 6 men 22–25 years old Random selection	6 professional sportsmen from the National Team of the Polish Judo Association 6 men 22–25 years old Random selection	A series of 15 therapy sessions was conducted for the experimental group, after which the authors introduced a break under a modified Michael Thompson Training, who is the author of the basic concept of applied psychophysiology. Each was preceded by a 3 min EEG diagnosis using a single-channel lead, and reference leads are also needed. Diagnostics were performed with eyes open and closed and with an activation task. During the task, the reference electrode was attached to the left ear, the ground electrode to the right ear, and the active electrode to the Cz point, while the active electrode was repositioned to the C3 point during training. The signals were filtered within the range of 2–40 Hz. Periods without artifacts were analysed. The testing protocol followed was beta1/theta, which is used to increase concentration and narrow the attention of the athletes. The control group underwent the same training procedure. EEG simulation was displayed independent of the subject’s wave patterns instead of the difference regarding the implementation of the beta1/theta protocol. The study was carried out with the Biograph Infiniti 6.0 and using the ProComp 5 decoding device, which was equipped with a 5-channel lead and an EEG sensor. The prerequisite for starting the EEG diagnosis was an impedance below 5 kΩ, and the measurement between the electrodes had to be greater than 1 kΩ. Respondents were advised to refrain from using medications and stimulants 12 h before each training session and test. Visual reaction speed was tested by a computer system using the Vienna Test System (WST).	Data taken for interpretation in the control group after the first and second cycles of testing for the group showed no statistically significant changes. In contrast, in the experimental group, a significant linear decrease in theta waves and an increase in beta waves were observed after the first and second training cycles.	Neurofeedback training caused significant changes in the preparation of athletes for sports competitions. In studies, it has been reported that the greatest therapeutic effects, in terms of reaction time training, were achieved between the fourth and fifth weeks of training. The study was conducted among a fairly small group of athletes, but it was a homogeneous group with similar ages, representing similar levels of training. The applied EEG diagnostics allowed for more precise results and improved the quality of evidence from the study. The authors themselves pointed out that there is little research on the changes in EEG wave dynamics in relation to individual training sessions. Neurofeedback training has measurable benefits; however, the methodology for conducting the training is still unsystematic and needs to be further explored.
Fatemehsadat Hosseini et al. [25], clinical study	15 elite and 15 non-elite volleyball players		Neurofeedback training was based on increasing the amplitude of the sensory motor band. Volleyball players’ EEG recordings were collected using the ProComp Infiniti. One training session lasting 30–45 min.		Increase in serving efficiency after neurofeedback training, especially volleyball players from elite groups.
Mikicin Mirosław et al. [26], clinical study	17 students of the Military University of Technology, who are professional soldiers and attend additional shooting classes	10 students of the Military University of Technology, who are professional soldiers and attend additional shooting classes	In both groups, the aim was to strengthen the beta frequency. In the experimental group, the subjects received positive or negative feedback only on the computer screen. In the control group, feedback was unrelated to the actual actions of the subjects. The training was provided using the EEG DigiTrack Biofeedback system. The average testing period was about 90 min, and 1 neurofeedback training session lasted about 40 min, in 20 sessions which took place 1–2 times a week.		For professional soldiers, level of attention and arousal are important skills. Both improved significantly in the experimental and control groups.
Cheraplina Larisa [27], clinical study	321 persons from different sports, elite athletes		At first, the “background” cerebrum bioelectric activity of all examined persons (n = 321) was registered in trials with open and closed eyes, with a following analysis of absolute and relative power in frequency bands of theta (4–8 Hz), alpha (8–13 Hz) and beta (13–20 Hz) activity. A second-stage neurofeedback course was directed towards increasing EEG power in the alpha-band via Pogoda Eva’s methodology. At the first stage, the duration of each trial was 5 min. During the second stage, 217 sportspersons had a 15-day neurofeedback course. The sessions were held once a day, continuing for 25–30 min.		The conducted research allowed concluding that the revealed changes in the sportsmens’ EEG depend on the kinematic characteristics of executed movements, sport skill and sex. These factors influence neurofeedback success but only in aggregation.
Mikicin Mirosław et al. [28], clinical study	7 elite swimmers		The athletes participated in the pre-test (Test 1), followed by the Neurofeedback training sessions (on a swimming ergometer), and the pre-test was performed after completion of all the sessions (Test 2). The following characteristics were measured in Test 1 and Test 2: anthropometric indices and body composition. Additional tests were performed: Kraepelin test, Holter EEG and EMG during 10 min of physical exercise, Progressive Test and the Wingate Test. Each exercise was performed on different days in the morning. Training sessions for 4 months. The study participants performed 20 neurofeedback training sessions (every 7 days, on average) using a swimming ergometer, with a continuation of conventional swimming training.		Neurofeedback-EEG training had an effect on the optimisation of psychomotor activities. Neurofeedback training during physical exercise may suggest the tendency towards maintaining energy and consistency in action.
Mikicin Mirosław et al. [29], clinical study	25 athletes (15 males and 10 females), student athletes involved in swimming, fencing, track-and-field, taekwondo, judo	10 athletes (5 males and 5 females)	Training consists of autogenic, audio–visual relaxation with eyes closed after everyday athletic training (in home conditions). The EEG examination (in resting supine position with eyes open and closed), attention reaction test and addition test for evaluation of Kraepelin’s work curve were carried out on each subject at the beginning and the end of the NFB training. The experimental group participated in 20 sessions of neurofeedback training for 4 months (every 7 days, on average). The control group performed regular sport training for 4 to 7 months between EEG recordings, similarly to sportspersons from the experimental group, but without parallel neurofeedback training and relaxation sessions.		The visual–neurofeedback training and audio–visual alpha relaxation training constitute holistic assistance in athletic training. They produce changes manifested in functionally different, eyes-open and eyes-closed states of the brain.
Maszczyk Adam et al. [30], clinical study	6 male athletes from the National Team of the Polish Judo Association	6 male athletes from the National Team of the Polish Judo Association	The NFB training protocol was performed and recorded using the Deymed Truscan system with 24 active channels. The effect of NFB training was examined by computer-based simple and complex reaction tests and selected tests of the Vienna Test System (VST). The research was carried out in 2 cycles. The first cycle included 15 training sessions held every other day. The duration of a training session was 10 min. The second part of the research, which took place after a 4-week interval, was characterised by the same frequency of training sessions, but at a reduced duration (4 min).		There were statistically significant differences between the control and experimental groups in the results of reaction speed tests after individual cycles of training. the influence of the frequency and duration of training on the result is demonstrated.
Balconi Michela et al. [31], clinical study	50 Italian subjects with valid driver’s licences participated in the study (38 females, 12 males)		Pre- and post-treatment assessment included behavioural, psychometric, neuropsychological and psychophysiological autonomic measures. The Driver Behavior Questionnaire (DBQ) and the Active Box (AB) device were used to evaluate everyday driving behaviour. Lasting 21 days and including daily sessions for practice (gradually incremented duration: from 10 min a day to 20 min per day).		The results allowed to underline improvement in driving behaviour performance and a decrease in behind-the-wheel violations among the experimental group. They displayed a physiological, behavioural and neuropsychological increase in efficiency related to attention, as well as a driving-related behavioural improvement after NF training.

**Table 4 brainsci-13-00660-t004:** PEDro scale for the included studies.

Reference	1	2	3	4	5	6	7	8	9	10
Nika Pusenjak et al. [14]	-	-	+	-	-	-	+	+	+	+
Camille Jeunet et al. [15]	-	-	-	-	-	-	-	+	+	+
Magdalena Krawczyk et al. [24]	+	+	+	+	+	+	+	+	+	+
Fatemehsadat HOSSEINI et al. [25]	-	-	-	-	-	-	+	+	+	+
Mikicin Mirosław et al. [26]	+	+	+	+	+	+	+	+	+	+
Cheraplina Larisa [27]	-	-	-	-	-	-	-	+	+	+
Mikicin Mirosław et al. [28]	-	-	-	-	-	-	-	+	+	+
Mikicin Mirosław et al. [29]	+	+	+	+	+	+	+	+	+	+
Maszczyk Adam et al. [30]	+	+	+	+	+	+	+	+	+	+
Balconi Michela et al. [31]	+	+	+	-	-	-	+	+	+	+

PEDro items: 1. Randomisation; 2. Allocation concealment; 3. Comparability at baseline; 4. Patient blinding; 5. Therapist blinding; 6. Assessor blinding; 7. At least 85% follow-up; 8. Intention to treat analysis; 9. Between-group statistical comparisons; 10. Point measures and measures of variability. Marks: (+), item fulfilled; (-), item not fulfilled.

**Table 5 brainsci-13-00660-t005:** EEG neurofeedback advantages in relation to sports training.

Neurofeedback in Sports Training
Support in convalescence
Dealing with failure better
Increase reflexes
Coordination improvement
Concentration improvement
Strengthening psychophysical resistance
Mood and self-esteem improvement
Increase in motivation
Better emotional control
Faster achievement of deep relaxation state

## Data Availability

The data presented in this study are available on request from the corresponding author.

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
