# Peer review of "The Use of Neurofeedback in Sports Training: Systematic Review"

_brainsci, 2023, doi:10.3390/brainsci13040660_

Round 1

Reviewer 1 Report (Previous Reviewer 3)

"The use of the neurofeedback in sports training. Literature review"

The author re-presented the paper with a modified title. The word "EEG biofeedback" was changed to "Neurofeedback". The authors are invited to provide justification for this alteration.

Figure 1. Diagram of bioofeedback functioning (own diagram).

The diagram is unclear, especially in regards to the label frequency adjustment as it is not clear what it is referring to.

"Depending on the needs of a particular subject, particular modules are safe, effective and bring the desired effect. "

Statement unclear

Line 102-115

The definition of Motor Imagery appears to be out of context, with no apparent connection to the preceding statements or to the potential benefits of Motor Imagery and/or Motor Imagery and Action Observation in sports.

Line 122

NFT undefined

In my previous review, I identified issues in the introduction, such as incorrect citations and unclear statements. There are numerous conceptual errors in the text, and some paragraphs appear disconnected from each other. For instance, it is unclear how the mentioned EEG waves relate to neurofeedback. The inclusion of motor imagery is not properly explained. In my opinion, the introduction requires significant revision to meet the standards for publication. The current version is difficult to comprehend and lacks proper documentation.

Discussion

The discussion section exhibits inconsistency in terminology usage. The authors initially discuss neurofeedback in the introduction but interchange it with EEG biofeedback in the discussion without providing a clear definition of either term. Furthermore, the discussion section is excessively long and challenging to comprehend. The inclusion of other types of biofeedback further complicates the discussion.

After careful evaluation of the article, I respectfully propose that it be rejected due to significant errors, particularly in the Introduction and Discussion sections. The inconsistencies in terminology usage, improper citations, and lack of clarity in the text render it unsuitable for publication. Furthermore, the discussion section is excessively long and challenging to comprehend, with a mixture of other types of biofeedbacks that further complicates the discussion. I suggest that the authors undertake extensive revisions to these sections to bring the article up to the standards required for publication.

Author Response

Dear Reviewer,

Thank you very much for your time and valuable comments, which all have been considered and incorporated. The detailed list of responses is given below. We hope that the modifications and explanation will be acceptable for you.

Yours sincerely,

Rydzik, corresponding author

The author represented the paper with a modified title. The word "EEG biofeedback" was changed to "Neurofeedback". The authors are invited to provide justification for this alteration.

A:Neurofeedback, called electroencephalographic (EEG) biofeedback, and occasionally referred to as neurotherapy. Actually, Neurofeedback is a kind of biofeedback, which teaches self-control of brain functions to subjectsby measuring brain waves and providing a feedback signal.

Figure 1. Diagram of bioofeedback functioning (own diagram).

A: We have created a new diagram

The diagram is unclear, especially in regards to the label frequency adjustment as it is not clear what it is referring to.

"Depending on the needs of a particular subject, particular modules are safe, effective and bring the desired effect. "

A: Depending on the need, neurofeedback is a safe and effective method that brings the desired effect and is suitable for all age groups. We have created a new diagram

Statement unclear

Line 102-115

A: This has been corrected 

The definition of Motor Imagery appears to be out of context, with no apparent connection to the preceding statements or to the potential benefits of Motor Imagery and/or Motor Imagery and Action Observation in sports.

A:In general, in NF applications, brain signals are recorded with different neuroscientific methods [electroencephalography (EEG), magnetoencephalography (MEG), functional magnetic resonance imaging (fMRI), near-infrared spectroscopy (NIRS)], processed in real time by a computer and fed back to the NF user online via visual, auditory, and/or tactile feedback. Motor imagery of movements is used as mental strategy in neurofeedback applications to gain voluntary control over activity in motor areas of the brain.  Motor imagery, defined as mental simulation of a kinesthetic movement [15], can also modulate activities in the sensorimotor cortex without any physical movements of the body and often used as mental strategy in NF applications [16]. MI recording technique captures signals based on a user’s imagination of performing a specific task for example limb movement but without actually moving that limb. The imagination of moving a unilateral limb causes a variation of activations in a specific cortex area[20], which are translated into electrical signals propagated by volume conduction through multiple brain tissue. Motor imagery has been used to enhance athletic performance [21].

Line 122

NFT undefined

A: Neurofeedback (NFB) is a method that assists subjects to control their brain waves consciously. In fact, the electroencephalography (EEG) is recorded during the neurofeedback treatment [4].

In my previous review, I identified issues in the introduction, such as incorrect citations and unclear statements. There are numerous conceptual errors in the text, and some paragraphs appear disconnected from each other. For instance, it is unclear how the mentioned EEG waves relate to neurofeedback. The inclusion of motor imagery is not properly explained. In my opinion, the introduction requires significant revision to meet the standards for publication. The current version is difficult to comprehend and lacks proper documentation.

A:This section has been revised. In addition, the entire text was proofread by a native speaker, as evidenced by the attached certificate. Electroencephalographic (EEG) biofeedback therapy utilizes brain waves as feedback information. In EEG biofeedback therapy, data is collected using EEG, and the principles of biological control theory are used to obtain feedback regarding psychological and physiological processes.

Discussion

The discussion section exhibits inconsistency in terminology usage. The authors initially discuss neurofeedback in the introduction but interchange it with EEG biofeedback in the discussion without providing a clear definition of either term. Furthermore, the discussion section is excessively long and challenging to comprehend. The inclusion of other types of biofeedback further complicates the discussion.

A: The entire discussion section was written from the beginning

After careful evaluation of the article, I respectfully propose that it be rejected due to significant errors, particularly in the Introduction and Discussion sections. The inconsistencies in terminology usage, improper citations, and lack of clarity in the text render it unsuitable for publication. Furthermore, the discussion section is excessively long and challenging to comprehend, with a mixture of other types of biofeedbacks that further complicates the discussion. I suggest that the authors undertake extensive revisions to these sections to bring the article up to the standards required for publication.

A: Thank you again for your valuable comments, we have made a great many improvements we hope you will find the changes satisfactory

Reviewer 2 Report (Previous Reviewer 1)

The authors did improve the manuscript by addressing several aforementioned issues. However, 1) some remain unsolved and 2) there are other concerns that need to be resolved.

Major flaws:

-       The assessment of study quality has been added in the draft (line 175-185). It appears that the authors used the PEDro scale to complete this assessment. However, 1) Table 5 should be numbered as table 4 and added in the main body, not at the end of this study. 2) It is confusing why the authors mentioned the Cochrane Risk of Bias tool in the main text, but no results were reported. 3) The manner of conducting this assessment needs to be addressed. 4) Several included studies are without a control group (such as ref 14, 26, 28, 29, 32), please comment on how the scores were given to items 1, 2, 4, 5, 6, and 9, which need a control group to be fulfilled in the PEDro scale. 5) The results of this assessment were not addressed in the Results section.

-       The Discussion part is hard to follow. Can the authors use sub-titles to emphasize the logic here?

-       Please add discussions related to the limitations of this study as well as of the included studies.

Others:

-       Line 193 in the Results, this study should be a literature review instead of a systematic review.

-       Still, tables and figures are not indicated in the main text.

-       References are missing, such as lines 93, 99, 182, 194, 205, 215, 227, 234, 242, 263, 291, 300, 308, 309, 313, 318, 328, 341, 353, and 371 in the Discussion.

Author Response

Dear Reviewer,

Thank you very much for your time and valuable comments, which all have been considered and incorporated. The detailed list of responses is given below. We hope that the modifications and explanation will be acceptable for you.

Yours sincerely,

Rydzik, corresponding author

The authors did improve the manuscript by addressing several aforementioned issues. However, 1) some remain unsolved and 2) there are other concerns that need to be resolved.

A: Thank you again we have made the corrections. In addition, the work has been checked by a natve speaker

Major flaws:

-       The assessment of study quality has been added in the draft (line 175-185). It appears that the authors used the PEDro scale to complete this assessment. However, 1) Table 5 should be numbered as table 4 and added in the main body, not at the end of this study. 2) It is confusing why the authors mentioned the Cochrane Risk of Bias tool in the main text, but no results were reported. 3) The manner of conducting this assessment needs to be addressed. 4) Several included studies are without a control group (such as ref 14, 26, 28, 29, 32), please comment on how the scores were given to items 1, 2, 4, 5, 6, and 9, which need a control group to be fulfilled in the PEDro scale. 5) The results of this assessment were not addressed in the Results section.

Results from the assessment of risk of bias are summarized in Table 4. 10 studies were rated as having low risk of bias, and thereby considered for inclusion in quantitative analysis.

A: Thank you for your comment, we apologise for our error all articles have been re-verified and properly graded on the scale. We have also added tables to the main section and discussed the score. We hope that the changes will be acceptable to you

-       The Discussion part is hard to follow. Can the authors use sub-titles to emphasize the logic here?

A:This section has been revised. The entire discussion has been written from the beginning and the work verified by a native seaker

-       Please add discussions related to the limitations of this study as well as of the included studies.

A: This has been corrected

 Limitations of the study

Two researchers performed the literature search, study selection, quality assessment, and data extraction. There were a small number of studies included in this review, which suggests a bias in the interpretation of the results. Most studies lacked a control group, making it difficult to determine whether effects are due to neurofeedback interventions or due to non-specific effects. Another limitation is that studies have used different neurofeedback protocols with different waves and frequency.

Others:

-       Line 193 in the Results, this study should be a literature review instead of a systematic review.

A: This has been corrected

Qualitatively summarizes evidence on a topic using informal or subjective methods to collect and interpret studies

-       Still, tables and figures are not indicated in the main text.

A: Added.

-       References are missing, such as lines 93, 99, 182, 194, 205, 215, 227, 234, 242, 263, 291, 300, 308, 309, 313, 318, 328, 341, 353, and 371 in the Discussion.

[4]. [14]. [23]. [10]. (25). Line 194and 205 in results section,

Round 2

Reviewer 2 Report (Previous Reviewer 1)

This manuscript has been improved substantially. Two suggestions:

-       The results of the assessment of study quality need to be mentioned in the Results. Hence, lines 210-211 in Method should be removed from the Method and put into the Results section, and Table 3 needs to be placed after Table 4.

-       There is a blank column in Table 3.

Author Response

Dear Reviewer, 

Thank you very much for your contribution to us manuscript and for taking the time to review it. We have re-done all your revisions. We hope it will be acceptable to you. 

Your Sincerly,

Łukasz Rydzik 

  •       The results of the assessment of study quality need to be mentioned in the Results. Hence, lines 210-211 in Method should be removed from the Method and put into the Results section, and Table 3 needs to be placed after Table 4.

A: This has been corrected 

-       There is a blank column in Table 3.

A: This has been corrected 

This manuscript is a resubmission of an earlier submission. The following is a list of the peer review reports and author responses from that submission.

Round 1

Reviewer 1 Report

This study adopted a literature review method to evaluate the effects of using EEG neurofeedback on improving the performance of professional athletes. However, there are several concerns regarding the methods, the results, and the manuscript structure that needs to be further addressed before the consideration of publication.

Major flaws:

-       There is a lack of novelty. A similar systematic review was published in 2017 (Mirifar A, et al. 2017).

-       To draw the targeted conclusion as stated in the manuscript, a systematic review or a meta-analysis should be considered. For doing this, I would suggest following the PRISMA 2020 statement (BMJ 2021;372:n71 | doi: 10.1136/bmj.n71).

-       Inclusion and exclusion criteria were not clearly stated. In the results section, it was stated that most single-unit studies were excluded, but this was not mentioned in the methods previously. Also, the exclusion criteria listed in table 2 did not match the statements in the main text.

-       It is not clear how many articles hit the research strategy during the first round, and the numbers of articles as well as the reasons for exclusion were not presented during each step of the literature screening process.

-       It seems like all the results were demonstrated in Table 3. However, most discussions appear to restate the results simply and redundantly in the main text.

Others:

-       Regarding the formality of academic publications, please pay attention to the usage of abbreviations and the preparation of the figures and tables. There is a lack of hints in the main text for indicating the number of tables and figures.

-       Please be aware of the accuracy of the statements. For instance, in the results section, the authors stated that “we excluded most of the articles that based their results on a single unit,” please define or give a certain number to indicate the so-called “most of” in the main text.

-       Critical references are missing from the main text, such as line 66-67 and line 96-98.

-       There is a need for the improvement of English writing.

Reviewer 2 Report

Title: The use of the EEG Biofeedback in sports training. Literature review

Article Type: review paper

Summary

In a systematic review paper, the authors have examined the effects of neurofeedback training on physical fitness of professional athletes representing various sport disciplines, such as judo, volleyball and soccer. Their results indicated that this type of practice has a significant effect on sport performance.

Evaluation

The topic of this study is interesting for publication in the Journal. The design for the study is appropriate to answer the research questions, and the paper is well written. For a review paper, the manuscript is quite straightforward and so I think that can be acceptable for publication. However, some points and suggestions should be addressed by the authors, in order to improve the quality of the manuscript.  

Minor points and suggestions

Please write about total number of the studies, the method of the review study, total number of the participants in all detected studies and also a better conclusion in the abstract.

EEG Biofeedback is neurofeedback, please change it through the manuscript, it means write neurofeedback instead of EEG Biofeedback in title and through the manuscript.

Why are other databases such as SCOPUS not checked? Please describe this in the manuscript.

What is the reason for choosing the period between 2012 and 2022?

What kind of the studies you included? Only RCT?

Due to the fact that the skill level can affect the how neurofeedback is effective, please mention the skill level of the participants in the studies in Table 3.  As I see, you mentioned for some studies, please write for all studies.

Please write the references correctly. The name of the publication is missing in some sources.  Please check all of them again exactly. For example, the correct way to write these sources is as follows.

51. Pourbehbahani Z, Saemi E, Cheng MY, Dehghan MR. Both Sensorimotor Rhythm Neurofeedback and Self-Controlled Practice Enhance Motor Learning and Performance in Novice Golfers. Behavioral Sciences. 2023 Jan;13(1):65.

48. Dobiasch M, Krenn B, Lamberts RP, Baca A. The Effects of Visual Feedback on Performance in Heart Rate-and Power-Based-Tasks during a Constant Load Cycling Test. Journal of Sports Science & Medicine. 2022 Mar;21(1):49.

Reviewer 3 Report

The paper titled "The use of EEG Biofeedback in sports training: Literature review" provides an insightful analysis of the application of EEG biofeedback in sports training. The paper presents a comprehensive literature review of the topic, highlighting the benefits and limitations of EEG biofeedback as a training tool.

The introduction of the paper provides a clear background on the concept of EEG biofeedback and its potential application in sports training. However, I have some remarks and comments:

"theta waves (4-8 Hz) that cause ineffective functioning of the human psyche "

The term "ineffective functioning" lacks clarity. It would be helpful to provide further explanation or clarification. Additionally, including references to support claims made in the paper would enhance its credibility and reliability.

Also,the terms "neurofeedback" and "EEG biofeedback" appear to be used interchangeably in the paper but are not adequately defined. It would be beneficial to provide clear definitions of both terms to avoid confusion and ensure proper understanding.

"The purpose of this was to increase alpha wave activity, which would translate 58 into an increased degree of relaxation By distinguishing different frequencies and their 59 roles, we can assign specific states to the appropriate waves associated, in the case of 60 gamma waves, with intense concentration of attention and assist the brain in processing 61 and combining information from different areas of the brain"

Neurofeedback and EEG biofeedback seem to be used interchangebly, but not explaiend well. Please provide definitions.

Missing references.

"Neurofeedback enables us to get better control over the physiological processes 64 taking place in the human body, which is difficult to control."

Missing references

"EEG biofeedback treatments are used 66 in medicine, where it has become a substitute for many pharmaceuticals."

Missing reference.

The same method is also common in geriatric treatments. 

Missing reference.

These patients often suffer 69 from various types of cognitive impairment resulting from damages in the central nervous system. 

Missing reference.

EEG-NFB training could be used according to the FITT principles (frequency, intensity, time and type) independent of age and fitness level.

Missing reference.

"In the 1980s, scientists developed a method of periodization of sports training."

Missing reference, unclear statement.

The commonly used concept of "biofeedback" should be associated mainly with the measurement and evaluation of various body parameters, with association with special electronic devices

Unclear definition. Please rewrite.

 EEG biofeedback training has the function of stimulating the athlete's body to self-regulate and create an autonomous impulse that can help in taking appropriate actions and decisions at key moments in the competition [7].

 It is not clear in what sense stimulating? 

 Achieving this state causes athletes to feel positive emotions and their heart rhythm and level of perceived stress areconstantlyable to better control [9]. 

 Please provide more references for the claim.

 Maximizing athletic form is ensured by proper motor and mental preparation. Fatigue, stress, and pressure disturb both aspects. Appropriate motor training 94 methods take care of the physical sphere, while mental training methods are techniques  that make the athlete's psyche more resistant to stress factors.

 Please provide references for the claims. 

It appears that the authors did not provide proper references to support many of the claims made in the introduction. The lack of references could potentially weaken the credibility and reliability of the paper's claims.

The authors have failed to address the definition of EEG-based motor imagery, motor activation during action observation, and their impact on motor performance in the introduction of the paper (please see 10.1007/978-3-030-31635-8_137 10.1016/bs.pbr.2017.08.008  10.1016/j.bandc.2021.105705 etc. 

Motor imagery, also known as mental practice, is a technique that involves creating vivid mental images of performing specific physical movements without actually physically executing the movements. In sports, motor imagery is used as a training tool to improve performance by enhancing motor skills, focus, and attention.

Research has shown that motor imagery can significantly improve sports performance. For example, studies have demonstrated that mental practice can enhance athletes' strength, endurance, and accuracy in sports such as basketball, tennis, and skiing. Furthermore, motor imagery has been found to be especially beneficial for athletes recovering from injuries, as it can help maintain muscle strength and prevent performance decline. 

Overall, motor imagery is a valuable tool that can improve sports performance by strengthening motor skills, enhancing concentration, and facilitating muscle memory. see 10.1007/978-3-030-31635-8_137

This omission may limit the reader's understanding of the paper's scope and objectives.

Discussion 

The benefits of this method have been documented in the papers around the world.

Weak claim. Please reference the papers and other similar reviews.

The subject may prove to be unable to control some of his brain parameters even after training. 

What brain parameters, unclear statement.

Rogala et al. believe that there is no significant 25 correlation between training intensity and NFB success. 

Reference missing, please define NFB before using it.

In many cases, the method of biofeedback and neurofeed- 30 back has contributed to enormous success.

Missing references. Which cases?

Pilot studies have demonstrated improved 31 performance and stress reduction in athletic athletes. 

Which pilot studies?

Upon further review, it appears that the article has several critical issues that need to be addressed before it can be considered for publication. These issues include missing references and unsupported claims, which significantly weaken the article's credibility and reliability.

Given the limited time given to the authors and the substantial work required to correct these issues, it is recommended that the article be rejected. However, if the authors are willing to invest the necessary effort to address the article's flaws, they may consider resubmitting the article for review after revision.

Additionally, the discussion section of the article is too lengthy and does not adequately reflect the paper's aspects. It would be beneficial to shorten the discussion section and ensure that it aligns with the study's objectives and findings.